# Disparities across Diverse Populations in the Health and Treatment of Patients with Osteoarthritis

**DOI:** 10.3390/healthcare9111421

**Published:** 2021-10-22

**Authors:** Warachal E. Faison, P. Grace Harrell, David Semel

**Affiliations:** 1Medical Affairs, Internal Medicine, Pfizer Inc., 235 E. 42nd Street New York, New York, NY 10017, USA; david.semel@pfizer.com; 2Department of Anesthesiology and Perioperative Medicine, Tufts Medical Center, 800 Washington Street, Boston, MA 02111, USA; pgraceharrell@gmail.com

**Keywords:** osteoarthritis, disparities, racial, ethnic, prevalence, treatment utilization, clinical outcomes, economic outcomes

## Abstract

The study of disparities across diverse populations regarding the health and treatment of patients with osteoarthritis (OA) is recognized as a priority for investigation and action by the National Institute of Arthritis and Musculoskeletal and Skin Diseases (NIAMS) and the American Academy of Orthopedic Surgeons (AAOS). OA is a common condition that increases with age, but with prevalence generally similar across racial and ethnic groups. However, disparities in the treatment of OA among racial, ethnic, and socioeconomic groups are well-documented and continue to rise and persist. The reasons are complex, likely involving a combination of patient, provider, and healthcare system factors. Treatment disparities among these different populations have an impact on clinical outcomes, healthcare, and productivity, and are projected to increase significantly with the growing diversity of the United States population. The aim of this short review is to summarize studies of racial, ethnic, and socioeconomic disparities among patients with OA in the United States, with a focus on prevalence, treatment utilization, and clinical and economic outcomes.

## 1. Introduction

Osteoarthritis (OA) is among the leading causes of pain and disability in the United States, with more than 27 million Americans living with OA [1]. In a more recent evaluation, it was that estimated 54.4 million people (22.7% of the population) reported doctor-diagnosed arthritis, and 23.7 million (43.5% of those with arthritis) had arthritis-attributable activity limitations [2]. OA is a significant cause of musculoskeletal pain, joint stiffness and swelling, and restricted range of motion [3]. These factors adversely impact upon mobility, activities of daily living, quality of life and psychological functioning [4,5,6,7]. Increasing annual costs, whether they be direct (i.e., medical expenditure) or indirect (i.e., loss of earnings), continue to contribute to significant burdens of OA on society [8].

Among the various risk factors, one that appears to contribute importantly to OA-related pain and disability is racial and ethnic background [9,10,11,12,13]. Disparities among these populations, including those related to patient factors, and differences in treatment offered by healthcare providers and access to healthcare systems, are well-documented and continue to rise and persist. Research in this area is particularly important given the myriad of publications, reports, and directives that have been issued in an effort to both draw attention to and reduce racial disparities. The National Institute of Arthritis and Musculoskeletal and Skin Diseases (NIAMS) and the American Academy of Orthopedic Surgeons (AAOS) have made treatment and health outcome disparities across diverse populations a major priority for investigation and action [14,15].

The aim of this short review is to summarize studies of racial and ethnic disparities among patients with OA in the United States, with a focus on disease prevalence, treatment utilization, and clinical and economic outcomes.

## 2. Prevalence of Arthritis/Osteoarthritis among Racial, Ethnic, and Regional Subpopulations in the United States

### 2.1. Prevalence of Arthritis by Race/Ethnicity

In 2017, the US Center for Disease Control (CDC) analyzed 2013–2015 data from the National Health Interview Survey (NHIS), an annual, nationally representative, in-person interview survey of the health status and behaviors of the noninstitutionalized civilian U.S. adult population, to update previous prevalence estimates of arthritis and arthritis-attributable activity limitations [2]. In each household identified, one family member was randomly selected to complete a “sample adult” questionnaire. Sampling weights were applied to account for household nonresponse and oversampling of non-Hispanic Black patients, Hispanics, and non-Hispanic Asians (Asians). Poststratification adjustments were applied by the National Center for Health Statistics (NCHS) using 2010 census estimates for the years 2013–2015. NHIS data from 2013 (*N* = 34,557), 2014 (*N* = 36,697), and 2015 (*N* = 33,672) were combined and weighted.



**Key insight** [2].

The results of the analyses showed that, in line with previous reports, White and Black patients had a similar age-adjusted % prevalence of doctor-diagnosed arthritis (22.6 (22.2 to 23.1) and 22.2 (21.4 to 23.0), respectively; Figure 1), but the prevalence of arthritis-attributable activity limitations (AAAL) was higher among Black patients. Hispanics and Asians had a much lower % prevalence of arthritis compared to White and Black patients (15.4 (14.6 to 16.1) and 11.8 (10.9 to 12.8), respectively; Figure 1), but a proportionately higher prevalence of AAAL. In contrast, multiple race (non-Hispanic) and American Indian/Alaskan Native populations had comparable % age-adjusted doctor-diagnosed prevalence of arthritis to White and Black patients. These data highlight that disparities in treatment utilization are not entirely based on prevalence of the disease, but more on other factors that will be discussed later in this review.

### 2.2. Prevalence of Knee OA by Race/Ethnicity and Age

The knees are among the joints most commonly affected by OA, and persons with knee OA commonly experience pain, aching, stiffness, and associated functional loss. Risk factors for the development of knee OA are multifactorial and include genetics, female sex, obesity, and knee injury [16]. People with advanced knee OA are thus at risk for undergoing total knee replacement (TKR), which, while effective, is an expensive procedure [17,18]. In 2016, Deshpande and colleagues performed analyses on data from the NHIS (2007–2008 and 2011–2012) to estimate the impact of several factors, including race/ethnicity and age, on the number of persons with symptomatic knee OA in the United States [19].



**Key insight** [19].

Results showed that in 2007–2008, approximately 14 million persons had symptomatic knee OA, with advanced OA comprising over half of those cases. The number of cases of patients with symptomatic knee OA rose to over 15 million in 2011–2012, with the number of people with advanced OA reaching nearly 9 million. The data showed that the prevalence of OA increased with age but was comparable across different racial/ethnic populations. In 2011–2012, the % prevalence of advanced disease in persons aged 45–64 years ranged between 2.9% for persons categorized as being from other racial background to 4.8% for Black patients; Figure 2 [19]. In persons aged 65 years and over, the % prevalence of advanced symptomatic disease increased to between 9.6% for persons categorized as being from other racial background to 12% for Black patients; Figure 2 [19].

### 2.3. Prevalence of Arthritis in Rural Versus Urban Communities

Rural populations in the United States have well documented health disparities, including higher prevalence of chronic health conditions [20,21]. Arthritis prevalence is known to vary widely by state (range = 19–36%) and county (range = 16–39%), but knowledge about the prevalence of arthritis and AAAL across urban–rural areas overall and among selected subgroups remains unclear [22]. In order to estimate the prevalence of arthritis and AAAL by urban–rural categories, the CDC analyzed data from the 2015 Behavioral Risk Factor Surveillance System (BRFSS) [23]. BRFSS is a state-based, random-digit–dialed landline and cellphone survey of the noninstitutionalized adult population aged ≥18 years of the 50 states, the District of Columbia (DC), and the U.S. territories; complete information was obtained from 426,361 individuals. Counties were classified into six urban–rural categories using the National Center for Health Statistics 2013 Urban–rural Classification Scheme for Counties and ranged from large central metropolitan (city) to noncore (rural). Unadjusted overall, age-specific, and age-standardized prevalence was estimated for arthritis and AAAL by urban–rural categories. Age-standardized prevalence by urban–rural categories was further stratified by selected demographic (sex, race/ethnicity, highest education level, and employment status) and health (body mass index, leisure time physical activity, self-rated health, disability, and smoking status) characteristics.



**Key insight** [23].

Analyses showed that the %prevalence of age-standardized doctor-diagnosed arthritis was higher in rural versus urban areas for White patients (27.7 (27.0 to 28.4) and 21.4 (20.8 to 22.0), respectively), Black patients (25.8 (23.8 to 27.9) and 22.9 (21.7 to 24.1), respectively), Asian (23.5 (16.6 to 32.3) and 11.4 (9.4 to 13.6), respectively), and multiracial, non-Hispanic populations (35.1 (29.8 to 40.7) and 24.6 (21.2 to 28.3), respectively) (see Figure 3) [23]. In contrast, the distinction between rural versus urban prevalence of arthritis was not present in Hispanic or American Indian/Alaskan Native populations (see Figure 3) [23].

## 3. Disparities in Healthcare Utilization among Patients with Arthritis/Osteoarthritis

### 3.1. Treatment Utilization: Total Joint Arthroplasty (TJA)

Total knee arthroplasty (TKA) and total hip arthroplasty (THA) are two of the most common major surgical procedures performed in the United States [24]. Racial disparities in primary TKA/THA have been well documented, with studies showing reduced utilization of surgery and higher complication rates in Black patients compared to White patients [25,26,27,28]. While some studies have been conducted within discrete healthcare systems at a single time-point, a pivotal analysis by Jha and colleagues found that racial disparities in TJA utilization had increased between 1992 and 2001 [25,26,28]. More recently, Singh and colleagues performed an analysis over an 18-year period (1991–2008) [29]. Data were retrieved from Medicare Part A (MedPAR) to identify four separate cohorts of patients (primary TKA, revision TKA, primary THA, revision THA). For each cohort, standardized arthroplasty utilization rates were calculated for White and Black Medicare beneficiaries for each calendar year and changes in disparities were examined over time. In 1991, utilization of primary TKA was 36% lower for Black compared to White patients (20.6 per 10,000 for Black; 32.1 per 10,000 for White; *p* < 0.0001); in 2008, utilization of primary TKA was 40% lower for Black patients (41.5 per 10,000 for Black; 68.8 per 10,000 for White; *p* < 0.0001) with similar findings for the other cohorts [29]. During the study period, standardized utilization rates for both primary and revision THA were significantly lower for Black patients as well [29]. See Figure 4 for 2008 THA and TKA utilization rates for Black and White patients [29].

Similar results were found in a more recent study by MacFarlane and colleagues where data from the Vitamin D and Omega-3 Trial (VITAL) including 25,874 adults (20% of whom were Black) were analyzed [30]. A subgroup of patients highly likely to have knee OA based on severity of knee pain, physician-diagnosed knee OA, and inability to walk short distances without pain was identified. Self-reported incident TKA was examined annually in follow-up (median: 3.6 years), adjusting for demographic and socioeconomic characteristics, comorbidities, and WOMAC pain and function. Proportionally, fewer Black than White patients had TKA (11% vs. 19%, *p* < 0.001); the cumulative incidence of TKA was also lower in Black participants than in White participants. The incidence rate (IR) per 100 patient-years for initial TKA was lower for Black than for White patients (2.5 (95% CI 1.5, 3.4) versus 4.3 (95% CI 3.5, 5.0), respectively), as was the IR per 100 patient-years for secondary TKA (on contralateral knee) (0.9 (95% CI 0.3, 1.5) versus 2.0 (95% CI 1.5, 2.5)). The overall TKA IR per 100 patient-years was also lower for Black participants compared with Whites (3.2 (95% CI 2.1, 4.4) versus 5.8 (95% CI 4.9, 6.7), respectively) [30].

Racial and ethnic disparities in treatment utilization are not only present in Black versus White OA patients, but also extend to other diverse populations. A study conducted by Zhang and colleagues aimed to study racial/ethnic disparities in the utilization of TKA in White, Black, Hispanic, Asian, Native American, and mixed-race individuals [31]. The authors analyzed data from over an 8-year period in eight racially diverse states using data from the State Inpatient Databases (SID). It was shown that, in comparison with whites (4.65 per 1000 population per year), TKA utilization rates were lower in Black (3.90), Hispanic (3.71), Asian (3.89), Native American (4.40), and mixed-race (3.69) populations and remained lower after adjustment for risk. Rates of TKA utilization for Black, Hispanic, and mixed-race groups became worse over time. Patients from minority groups were less likely to undergo TKA in high-volume hospitals than were Whites and the rates of mortality were significantly higher in racial/ethnic diverse populations, as were rates of complications following surgery.

Another study conducted by Cavanaugh and colleagues [31] investigated racial and ethnic disparities in TKA utilization in older White (*N* = 90,420), Black (*N* = 8942) and Hispanic (*N* = 3405) women in relation to demographic, health, and socioeconomic status variables [32]. Data were retrieved from participants enrolled in the Women’s Health Initiative (WHI), a prospective study that recruited post-menopausal women from across the United States [33,34]. Absolute disparities were determined using utilization rates by racial/ethnic groups and relative disparities were quantified using multivariable hazards models adjusting for age, arthritis, joint pain, mobility disability, body mass index, number of comorbidities, income, education, neighborhood socioeconomic status (SES), and geographic region. TKA utilization was higher among White women (10.7/1000 person-years) compared to Black (8.5/1000 person-years) and Hispanic women (7.6/1000 person-years). After adjusting for age, Black and Hispanic women with health indicators for TKA (e.g., diagnosis of arthritis, moderate to severe joint pain, and mobility disability) were significantly less likely to undergo TKA (Black: Hazard ratio (HR) (95% CI): 0.70 (0.63–0.79; Hispanic: HR: 0.58 (0.44–0.77)) compared with White women (HR: 1.0, reference value).



**Key insight** [35,36,37].

### 3.2. Patient Factors That May Influence Racial Disparity in TJA Utilization

Several patient-related factors have been identified that may influence racial disparity in TJA utilization among diverse populations. These include factors such as patient expectations, interaction and communication within social networks, knowledge of the procedure, cultural beliefs, and overall willingness to undergo the procedure (See Table 1) [25,38,39,40,41,42,43,44,45].

With respect to patient expectations, a study by Ibrahim and colleagues (*N* = 600; including elderly, male veterans with symptomatic knee or hip OA (or both)) showed that when asked, “How often do you think one dies from replacement surgery?”, Black patients were more likely than White patients to respond “sometimes or often,” (25% vs. 20%) [25]. Black patients were also more likely to believe that post-surgery care could last for more than 2 weeks (45% vs. 18%), and that the convalescent period would last for longer than 6 months (47% vs. 40%). Black patients also had lower expectations with respect to treatment outcomes and were more likely to anticipate moderate or extreme pain (62% vs. 42%) and moderate to extreme difficulty walking (64% vs. 39%) following TJA [25]. The findings related to lower expectations for successful treatment outcomes are reflected in a more recent study by Groeneveld and colleagues (*N* = 909; elderly male patients with moderate or severe OA of the hip or knee receiving primary care at 2 veterans affairs medical centers) [38]. Here, a well-validated survey instrument (Joint Replacement Expectations Survey) was used to determine if differences in expectations for pain relief, functional improvement, and psychological well-being following TJA could be explained by racial variation in disease severity, socioeconomic factors, literacy, or trust. Data showed that among potential candidates for TJA, African American patients had significantly lower expectations for surgical outcomes than White patients. For knee OA (*N* = 627), the unadjusted mean expectation score (scale 0–76) was significantly lower for African American patients compared to Whites (48.7 vs. 53.6; mean difference 4.9, *p*< 0.001). For hip OA patients (*N* = 282), scores (scale 0–72) were also lower for Black compared to Whites (45.4 vs. 51.5; mean difference 6.1, *p* < 0.001). Following multivariable adjustments, although differences were reduced, disparities were not entirely explained by racial differences in demographics, disease severity, education, income, social support, or trust.

Social interaction and communication within social networks are other factors identified that may influence treatment utilization among different racial and ethnic groups. A study conducted by Ibrahim and colleagues compared elderly African American and White patients with OA of the knee or hip with respect to their perceptions of the efficacy of traditional and complementary treatments and their self-care practices (*N* = 593 patients; 44% African American and 56% White) [39]. Between May 1997 and March 2000, male patients attending a primary care firm clinic associated with the Louis Stokes Cleveland VA Medical Center in Cleveland, Ohio, were randomly approached by trained interviewers. They were asked a series of questions based on the arthritis supplement of the National Health and Nutrition Examination Survey I in order to qualify for the study; questions were based on the presence, duration, and severity of hip or knee pain [39,46]. Patients who met the inclusion criteria were then asked a series of questions, including “If your hip/knee pain were to become severe, would you do the following?” More Black patients reported that they would “ask a friend or family for advice,” on whether they would go through with a TJA procedure. Cultural beliefs are sometimes overlooked as playing a part in treatment utilization differences among diverse populations. For example, studies have shown that Black patients are more likely to consider prayer as helpful to self-treat knee or hip pain compared to Whites [39,41]. These data highlight that the disparities in treatment utilization should take into account the reliance on self-care elements in the management of OA in certain racial and ethnic groups.

It is thought that racial and ethnic personal preferences may contribute to marked disparities in the utilization of TJA, and that Black patients are often influenced by personal and community knowledge of the procedure. A study conducted by Kwoh and colleagues was designed to identify the determinants of knee OA patients’ preferences regarding TKR by race and to identify the variables that may mediate racial differences in willingness to undergo surgery (*N* = 514, White; *N* = 285, Black) [40]. Findings showed that determinants of patient preference for TKR differed between Black and Whites. For Black patients, better understanding of the procedure, a shorter hospital course, less post-surgical pain and walking difficulty, favorable expectations of surgery, and trust in physicians significantly influenced patient willingness to undergo TKR. On the other hand, favorable expectations of surgical outcomes, trust in healthcare discussions with a physician, and not having received surgical referral were significant predictors of willingness among Whites. After adjustment for recruitment site, sex, age, income, WOMAC total score, health insurance, and social support, the odds of willingness to undergo TKR was 57% lower in Black compared to White patients (OR: 0.43; 0.28 to 0.67). These findings, in relation to Black patients’ willingness to undergo surgery, were confirmed in studies conducted by Allen et al., Ibrahim et al. Vina et al. and Hausmann et al. [42,43,44,45]. Overall, personal preferences and community knowledge of surgical procedures were generally perceived as negative by Black patients. These perceptions may play a part in the reluctance of Black patients to even consider undergoing an invasive corrective solution.

### 3.3. Healthcare Provider Factors That May Influence Racial Disparity in TJA Utilization

Several factors related to healthcare providers have been identified that may influence racial disparity in TJA utilization; a summary of these factors is shown in Table 2.

Efficient and informed communication between specialized surgeons and OA patients is an important factor in helping patients make constructive decisions regarding potential surgical interventions. Effective discussions are key to the delivery of the best quality of care and overall patient satisfaction; understanding any differences in this complex relationship is essential. Levinson and colleagues performed a study where they analyzed recordings of office visits between orthopedic surgeons and elderly White versus African American OA patients [47]. Even though interviews addressed similar questions, differences in the way the surgeon engaged with the patient and patient satisfaction ratings were clearly present; Black patients were less satisfied with the overall communication. Ratings related to responsiveness, respect, and listening in visits with Black patients were also lower compared with White patients.

Referral bias in favor of White patients is also thought to play a part in Black patients being less likely to receive a recommendation for TKA. A study conducted by Hausmann and colleagues examined whether orthopedic surgeons are less likely to recommend TJR to African-American patients (*N* = 120) compared to White patients (*N* = 337) with similar clinical indications [45]. Results were based on information gathered from patient surveys and surgeon notes. The data showed that the odds of receiving a TJR recommendation were lower for Black than White patients of similar age and disease severity (OR = 0.46, 95% CI = 0.26–0.83; *p* = 0.01). However, as discussed earlier, patient preferences for the procedure came into play and the difference was no longer significant when this factor was taken into account. In brief, following adjustments for patients’ pre-existing willingness to undergo TJR, the race difference in TJR recommendations decreased. This suggests that race differences in TJR recommendations may result from orthopedic surgeons being responsive to patient preferences regarding the procedure.

Cultural and psychosocial factors influence how patients experience and express pain. Another health provider-related factor thought to influence treatment utilization is that Black and White patients may use different descriptions for the quality of their knee or hip OA pain, which may lead to differential assessment by healthcare providers of the need for TJA [48]. When White and Black patients were asked to describe their pain based on a 4-factor model (factor 1 combined the variables “Sharp and Stabbing”; factor 2 combined the variables “Sore and Tender”; factor 3 combined the variables “Dull, Stiff, and Achy”; and factor 4 combined the variables “Hot, Frozen, and Throbbing”), differences did indeed exist. In addition, the relationship between quality of pain and global quality-of-life (QOL) ratings also varied between White patients and Black patients; patient descriptions of quality of chronic knee or hip pain did not correlate with radiologic stage of disease [48].

### 3.4. Healthcare System Factors That May Influence Racial Disparity in TJA Utilization

Healthcare system factors, including access and geographical variations, that may influence racial disparity in TJA utilization have also been identified (see Table 3), and it seems likely that racial and ethnic differences in financial constraints is one of these [49]. Hanchate and colleagues performed a study to estimate national TKA rates by economic factors, and the extent to which differences in insurance coverage, income, and assets contribute to racial and ethnic disparities in TKA use. Data were gathered from the US longitudinal Health and Retirement Study survey data for the elderly and near elderly (biennial rounds 1994–2004) from the Institute of Social Research, University of Michigan [50,51]. Analyses showed that TKA rates were notably lower for Black and Hispanic patients than for White patients, especially among Black men. After adjusting for demographic factors, illness burden and physical functional limitations, the deficits narrowed considerably, but the Black male deficit remained large (OR = 0.56). Findings highlighted that lower utilization of TKA among Black and Hispanic patients may be associated with insurance coverage limitations and unaffordable out-of-pocket costs [49].

Higher hospital surgical volumes have been associated with lower complication rates following TKA, with increased rates of mortality, pulmonary embolism and infection at low-volume hospitals compared to high-volume centers [52,53]. SooHoo and colleagues conducted a study to identify the characteristics of patients who undergo TKA at high-volume hospitals and their differences from those who receive care at low-volume hospitals [54]. Data showed that Black, Hispanic, and Asian/Pacific-Islander patients were significantly more likely than White patients to receive TKA at low volume facilities; Black patients (relative risk ratio (RRR) = 1.73, 95% confidence Interval (CI): 1.09–2.76, *p* = 0.02); Hispanic patients (RRR = 3.13, 95% CI: 2.31–4.23, *p* < 0.001) and Asian/Pacific Islanders (RRR = 2.95, 95% CI: 1.89–4.62, *p* < 0.001). Medicaid insurance was also an independent predictor of treatment at low-volume hospitals. This study supports the need to consider racial and socioeconomic disparities in efforts to improve the quality of care of patients undergoing TKA at lower-volume hospitals.

Another healthcare system factor that may influence racial disparity in TJA utilization is geographic regional variations in patient care. Regional variations in the rates of surgery are commonly considered to reflect differences in local medical opinion concerning the value of these procedures [27]. Skinner and colleagues used all Medicare fee-for-service claims data for 1998 through 2000 to determine the incidence of TKA according to Hospital Referral Region, sex, and race or ethnic group [27]. Data showed that, at the national level, the annual rate of TKA was higher for non-Hispanic White women (5.97 procedures per 1000) than for Hispanic women (5.37 per 1000) and Black women (4.84 per 1000). The rate for non-Hispanic White men (4.82 procedures per 1000) was higher than that for Hispanic men (3.46 per 1000) and more than double that for Black men (1.84 per 1000). The rates were significantly lower for Black men than for non-Hispanic White men in nearly every region of the country (*p* < 0.05).

### 3.5. Utilization of Other Treatments Apart from Surgery and Treatment Recommendations

There are a number of treatments available for patients who are not suitable for surgery or those that are not at a stage where surgery is deemed necessary. Two of these treatments include hyaluronic acid and/or corticosteroids which are injected directly into the joint. Lapane and colleagues performed a study to investigate the use of injections among adults with radiographically confirmed knee OA and identify factors associated with injection use [55]. Factors employed in the study included disease severity, race or ethnic group, and household income. Data gathered from the Osteoarthritis Initiative (OAI), an international collaboration sponsored by the National Institutes of Health (NIH) showed that economic and racial background did in fact influence injection use. Injection use was reported more commonly in patients with a high annual income and less commonly in Black patients (versus White patients), even after adjusting for clinical characteristics such as disease severity [55].

Another study performed by Yang and colleagues investigated differences between Black and White patients in using different treatment approaches (conventional versus alternative treatments) to manage symptoms among individuals with radiographic-confirmed knee OA [56]. Alternative treatments were defined into 7 categories: (1) alternative medical systems (e.g., acupuncture); (2) mind–body interventions (e.g., yoga, spiritual activities); (3) manipulation and body-based methods (e.g., massage, chiropractic); (4) energy therapies (copper bracelets, magnets); (5) biologically based topical therapies (lotions, capsaicin); (6) diet; and (7) supplements (e.g., herbs, vitamins). Using data gathered from the OAI, it was demonstrated that Black patients were less likely to use alternative treatments (either alone or in combination with conventional medications) versus White patients, but did show comparable use of conventional medications [56].

Physical therapy plays an essential role in the conservative management of symptoms experienced by individuals with knee OA and is recommended by the American College of Rheumatology and by other professional organizations [57,58,59]. Recent management guidelines recommend at least 6 months of conservative therapy, including physical therapy for patients with symptomatic knee OA prior to having surgery [60]. Iversen and colleagues performed post hoc analyses on baseline data from a randomized controlled trial of 350 adults with physician-diagnosed symptomatic knee OA to identify correlates of physical therapy utilization [61]. In this small cohort of patients, race itself was not a factor in physical therapy utilization, although sex, employment status, duration of knee symptoms, history of knee joint injection, and having family members with knee OA were. In another small observational study using patient questionnaires (*N* = 593), Ibrahim et al. showed that even though Black patients were more likely than White patients to seek advice from a physical therapist and were more likely to perceive physical therapy as helpful, actual use was not different between racial groups [39]. Racial disparity extends to healthcare provider recommendations for physical activity. Austin and colleagues performed a cross-sectional analysis of the Behavioral Risk Factor Surveillance System, 2011, 2013, 2015, (across 17 states) of patients ≥18 years of age with self-reported physician-diagnosed arthritis [62]. In this large survey sample (*N* = 83,376) of patients with self-reported physician-diagnosed arthritis, 6 in 10 patients received recommendations for physical activity from their health care provider, with Black and Hispanic patients being less likely than White patients to receive such recommendations [62]. The authors propose that unconscious stereotyping by healthcare providers, i.e., belief about individual patient’s ability to exercise, may have a played a role in the lower recommendation rate. However, it is important to note the limitations of this study, including possible patient recall bias as the data were self-reported, with variation in the length of time between last physician visit and survey participation.

## 4. Disparities in Treatment Outcomes for Patients with Arthritis/Osteoarthritis

### 4.1. Disparities in Treatment Outcomes: Total Knee/Hip Arthroplasty

Disparities in treatment outcomes for patients with arthritis/osteoarthritis across diverse populations are numerous and have been well documented. Despite this, they still exist, and continue to grow and persist. Black versus White patients have been shown to have longer hospital stays and higher readmission rates. They are less likely to be discharged to home and more likely to be transferred to a skilled nursing facility or another acute-care hospital. Black patients also experience higher complication rates following surgery (including morbidity and mortality), worse pain and function, and less overall satisfaction [29,31,63,64,65,66]. According to a systematic literature review of disparities in outcomes among Black versus White patients undergoing TKA, Black patients showed worse pain and function in four studies and less satisfaction in 1 study versus White patients (6 months to 2 years after surgery). This study also showed that disparities in treatment outcomes following TKA also extend to Native-American patients and mixed-race patients, with these groups experiencing higher mortality rates [31].

Another study, performed by Singh and colleagues, investigated whether racial differences influenced patient-reported outcome measures (PROMs) following primary THA and TKA between 2016 and 2020 [67]. Participants included 1999 THA patients and 1375 TKA patients. In the THA cohort, 1636 (82%) were Caucasian, 177 (9%) were African-American, and 186 (9%) were of other races. In the TKA cohort, 864 (63%) were Caucasian, 236 (17%) were African-American, and 275 (20%) were of other races. Results showed that compared to Caucasian patients, African-American patients demonstrated lower PROM scores following TJA. Surgical-time and length-of-stay significantly differed between the groups (*p* < 0.001); Black patients who underwent THA required the longest operative time and spent longer in hospital following THA/TKA compared to White patients [67].

In addition to disparities among different racial and ethnic groups following primary corrective surgery, different treatment outcomes may also exist following revision procedures. Roche and colleagues performed a study to evaluate, among others, the incidence and burden (defined as the ratio of revision to the sum of revision and primary procedures) experienced by diverse populations following revision TKA [68]. Data were gathered from the PearlDiver US Private Payer Database from between January 2007 and December 2014 (*N* = 125,901). Analyses showed that revision incidence and burden were the highest in the African-American cohort (12.4%, 11.1%), higher than that experienced by White patients (*p* < 0.001), and were lowest in the Asian cohort (3.4%, 3.3%) (*p* < 0.001).

Racial disparities are also reflected in estimates of quality-adjusted life years (QALYs) for patients with OA. A recent study performed by Kermann and colleagues used a validated computer simulation model (Osteoarthritis Policy Model) of the natural history and treatment of knee OA to estimate QALY in Black and White patients with and without TKA. According to the data, Black patients gain fewer QALYs (per 100 patients) from TKA compared with White patients (see Figure 5) [69].

### 4.2. The Influence of Poverty on Racial Disparity in Treatment Outcomes

Race is an important predictor of TKA and THA outcomes in the United States; however, socioeconomic factors can make analyses of the influence of race difficult and can confound identification of the root cause of disparate outcomes following surgery.



**Key insight** [66,69].

A study by Goodman and colleagues investigated whether race and socioeconomic factors at the individual level were associated with patient-reported pain and function, and if there was an interaction between race and community poverty and patient-reported pain and function 2 years after TKA [66]. Data obtained from hospital-based registries and census tracts showed that Whites and Black patients not considered to be living in financially challenging conditions had similar levels of pain and function 2 years after TKA (Western Ontario and McMaster Universities Osteoarthritis Index (WOMAC) pain, 1.01 ± 1.59 points lower for Black than for White, *p* = 0.53; WOMAC function, 2.32 ± 1.56 lower for Black than for White, *p* = 0.14). WOMAC pain and function scores worsened with increasing levels of community poverty, but did so to a greater extent among Black than White patients.

In a second study by Goodman et al., using a large institutional THA registry and census tract data, the authors analyzed interactions between race and percent of population with Medicaid coverage and its association with 2-year patient-reported outcomes following THA surgery [70]. Data showed that Black patients undergoing THA had worse baseline and 2-year pain and function scores compared with White patients, and that there were strong positive correlations between census tract Medicaid coverage and percent living below poverty. With respect to treatment outcomes 2 years post-surgery, Black patients in communities with high census tract Medicaid coverage had predicted WOMAC function scores over 5 points lower (80.42 versus 85.96) than those in less deprived communities, even among patients who had access to the same hospital care; this difference was not observed among White patients.

### 4.3. The Economic Impact of Racial Disparity in Treatment Utilization and Clinical Outcomes

Overall, it should be understood that if OA treatment disparities are not addressed, there may be significant increases in healthcare costs and productivity losses [71]. Karmarkar and colleagues used a Markov model to assess the economic burden of racial disparities in treatment utilization from the perspective of the patient, employer, and society [71]. The Markov model is an analytical framework that is frequently used in decision analysis, and is probably the most common type of model used in economic evaluation of healthcare interventions. Markov models use disease states to represent all possible consequences of an intervention of interest. Data showed that, from a patient perspective across treatment scenarios, Black women had generally higher lifetime costs (up to $91,974) compared with Hispanic women (up to $72,712) or White men (up to $85,093); estimates include both medical and non-medical (lost income) costs. From an employer perspective, Black women had higher productivity losses compared with White men after OA duration of 10 years; productivity loss was measured through reduction in income. From the perspective of society as a whole, the estimated 2015 total costs of the racial/ethnic treatment disparities in the care of knee OA (reflecting the product of the treatment utilization rate, lifetime cost of the pathway, and number of patients with OA in each population group) were $13.28 billion over a 40-year period; total costs were projected to increase to $15.6 billion in 2025 due to the growing diversity of the US population [71]. These data are clearly of concern, but projections of costs could be even higher when the limitations of the model used are taken into account. For example, data were estimated using a model built upon structural and parameter assumptions with inherent limitations; indirect costs only included lost income (and not others, e.g., caregiver or childcare costs, loss of leisure activities or retirement costs), and dynamic factors of disease progression were not considered.

## 5. Summary and Conclusions

It is clear from the data discussed that treatment disparities exist across diverse populations regarding the health and treatment of patients with OA in nearly all aspects of care. Although OA is a common condition that increases with age with a prevalence that is generally similar across racial and ethnic groups, disparities in the treatment of OA among these patients are well-documented and continue to rise and persist. Treatment disparities among these different populations have wide ranging consequences on clinical outcomes, healthcare, and productivity, and are projected to increase significantly with the growing diversity of the United States population.

Solving these challenges will not be straightforward; it will involve every stakeholder in the continuum of care, and will take time. The first step is to raise awareness of the treatment disparities that exist, and to challenge orthopedic surgeons and healthcare systems more generally to take action. From there, further analysis of the multifactorial reasons underlying disparity will be necessary to fully understand and to plan interventions—short and long term—that can address them. This will include activities such as increasing the current underrepresentation of Black and other minority orthopedic surgeons. According to the 2018 OPUS Survey [72], in 2018 only 1.9% of orthopedic surgeons were Black. Increasing representation of underrepresented minorities should be a key focus for medical schools and institutions. In this regard, initiatives such as “Nth Dimensions”, established by orthopedic surgeons in 2004 to address the underrepresentation of minorities in orthopedic surgery, have demonstrated success [73]. Implementing programs to improve referral pathways, which can be suboptimal in communities with higher concentrations of Black residents [74], would additionally be desirable.

Initiatives to address the negative impact of social determinants of health are perhaps a longer-term goal, but nonetheless important [75]. Collection of data on social needs via primary care settings, and implementation of nurse- or community-led programs, have been shown to considerably improve health outcomes (e.g., the Centers for Disease Control and Prevention’s Racial and Ethnic Approaches to Community Health (REACH) Programs) [76,77]. Removal of barriers to accessing health services (e.g., inadequate health insurance, transportation barriers) is an important step towards reducing health disparities [78].

Pharmaceutical and device companies can also be part of the solution—for example by increasing the diversity of investigators and trial populations in clinical trials [79]; by engaging with stakeholders on healthcare policy and access; and by helping to target education to areas and populations with the greatest need.

In conclusion, we must work together to find and implement solutions that reverse the rise in treatment disparities so that all patients with OA can benefit from appropriate treatments and experience a better quality of life.

## Figures and Tables

**Figure 1 healthcare-09-01421-f001:**
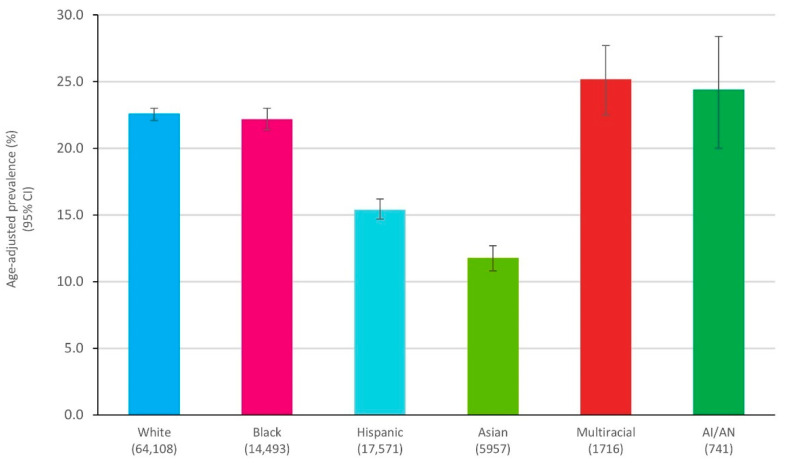
Prevalence of arthritis by race/ethnicity. Adapted from [2]. White, Black, Asian, Multiracial, and American Indian/Alaskan Native are non-Hispanic. Unweighted sample sizes are shown in parentheses. Abbreviations: AI/AN, American Indian/Alaskan Native.

**Figure 2 healthcare-09-01421-f002:**
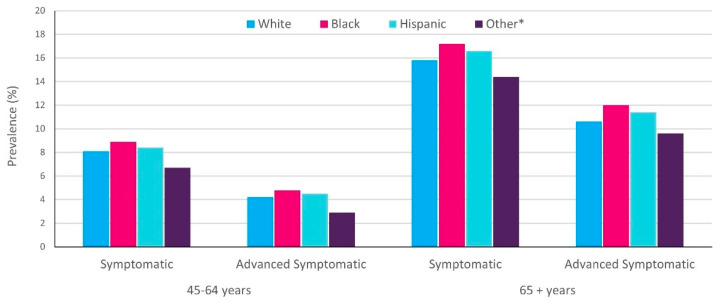
Prevalence of knee OA by race/ethnicity and age. Adapted from [19]. Data collapsed across males/females, with percentages recalculated. Data for 25–44 years are not shown as prevalence was generally consistent across groups and ranged from 1% (advanced symptomatic) to 2% (symptomatic). White, Black, Other are non-Hispanic. * American Indians/Pacific Islanders/Alaskan and Asian Americans.

**Figure 3 healthcare-09-01421-f003:**
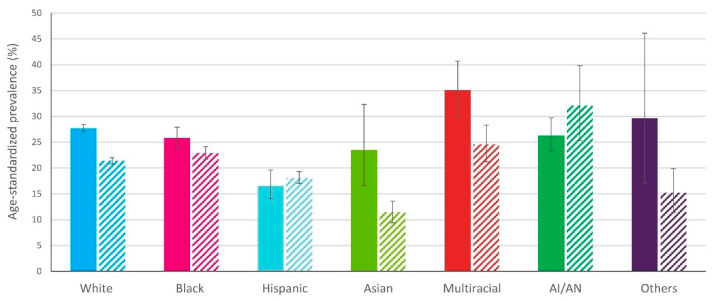
Prevalence of arthritis in rural versus urban communities. Adapted from [23]. Solid bars, rural; hatched bars, urban. White, Black, Asian, Multiracial, and American Indian/Alaskan are non-Hispanic. Rural refers to a noncore area; urban refers to large central metropolitan city. Abbreviations: AI/AN, American Indian/Alaskan Native.

**Figure 4 healthcare-09-01421-f004:**
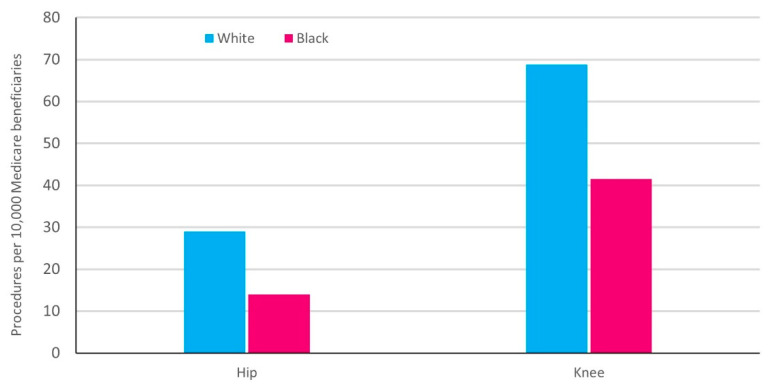
2008 Treatment utilization: Total joint arthroplasty. Adapted from [29].

**Figure 5 healthcare-09-01421-f005:**
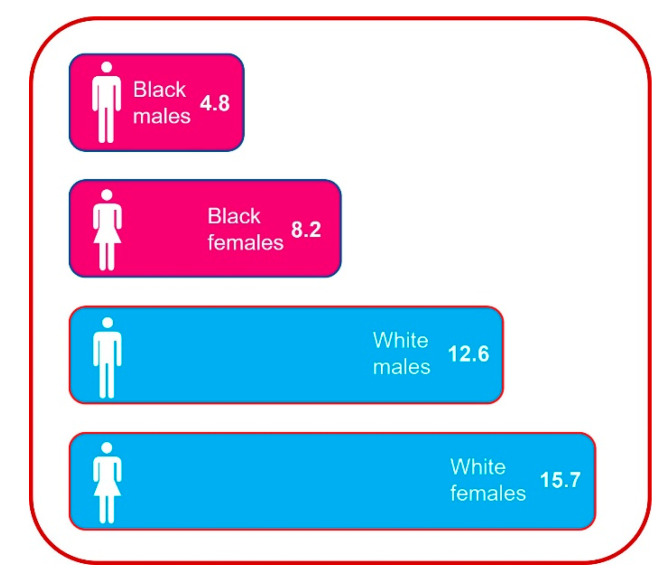
QALYs gained from TKA utilization per 100 patients. Adapted from [68]; Abbreviations: TKA, total knee arthroplasty.

**Table 1 healthcare-09-01421-t001:** Patient factors that may influence racial disparity in TJA utilization.

** Expectations **	• Black patients have **lower levels of expectation** for surgical outcomes
** Social Network **	• Black patients are **more likely to ask friends or family for advice** for severe OA pain
** Knowledge **	• When considering TJA, Black patients are **influenced by personal and community knowledge** of the procedure
** Cultural beliefs **	• Black patients are **more likely to consider prayer as helpful** to self-treat knee or hip pain
** Willingness/preference **	• Black patients are **less willing to consider or undergo TJA**

Unless otherwise stated, comparisons are with White patients. Abbreviations: OA, osteoarthritis; TJA, total joint arthroplasty. Sources: [25,38,39,40,41,42,43,44,45].

**Table 2 healthcare-09-01421-t002:** Healthcare provider factors that may influence racial disparity in TJA utilization.

** Communication **	• Black patients are **less satisfied with communication** with their orthopedic surgeon
** Referral bias **	• Black patients are **less likely to receive a recommendation for TKA**, but the effect is no longer significant after adjusting for patient preference for TKA
** Description of pain **	• Black and White patients may **use different descriptions for the quality of their knee or hip OA pain**, which may lead to differential assessment by healthcare providers of the need for TJA

Unless otherwise stated, comparisons are with White patients. Abbreviations: OA, osteoarthritis; TKA, total knee arthroplasty; TJA, total joint arthroplasty. Sources: [45,47,48].

**Table 3 healthcare-09-01421-t003:** Healthcare system factors that may influence racial disparity in TJA utilization.

** Access **	•**Insurance coverage and financial limitations** explain some of the racial or ethnic variations in total knee arthroplasty (TKA) rates • Black, Hispanic and Asian/Pacific-Islander patients are significantly more likely than White patients to **receive TKA at low volume facilities**
** Geographic **	•**Regional variations** may contribute in part to ethnic and racial disparities of TJA utilization - For example, in certain geographic regions, the TKA rate among Black women was significantly lower than that of White women, whereas rates were approximately equal in other regions

Abbreviations: TJA, total joint arthroplasty. Sources: [27,49,54].

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
