# Peer review of "Disparities across Diverse Populations in the Health and Treatment of Patients with Osteoarthritis"

_healthcare, 2021, doi:10.3390/healthcare9111421_

Round 1

Reviewer 1 Report

Well written and organized short review on a hot topic.

There is not much new information in this manuscript.  Despite, it may be worthwhile to publish this submitted manuscript as it is paying attention to a long standing problem which is seen in United States, in particular. If accepted, there two recommendations to be done. 1. Current literature is reviewed and summarized, but without criticism on research done by other authors. It seems to me that authors of this short review has outlined those conclusion that will fit to their conclusion. 2. However, conclusion is vague and very generalized. Here is example: "In conclusion, we must take a more focused approach to halt the rise in treatment disparities and then find solutions so that all patients with OA across diverse populations can benefit from appropriate treatments and experience a better quality of life.". To put in other words, this manuscript will be of more value, if authors make an additional effort to give specific and detailed guidelines on how to approach problem of disparities across diverse population in the health and treatment of patients with osteoarthritis.  Otherwise, this manuscript is only a summarized work done by other researchers.

Author Response

Reviewer 1 (Received September 5):  Well written and organized short review on a hot topic.

There is not much new information in this manuscript.  Despite, it may be worthwhile to publish this submitted manuscript as it is paying attention to a long standing problem which is seen in United States, in particular. If accepted, there two recommendations to be done. 1. Current literature is reviewed and summarized, but without criticism on research done by other authors. It seems to me that authors of this short review has outlined those conclusion that will fit to their conclusion. 2. However, conclusion is vague and very generalized. Here is example: "In conclusion, we must take a more focused approach to halt the rise in treatment disparities and then find solutions so that all patients with OA across diverse populations can benefit from appropriate treatments and experience a better quality of life.". To put in other words, this manuscript will be of more value, if authors make an additional effort to give specific and detailed guidelines on how to approach problem of disparities across diverse population in the health and treatment of patients with osteoarthritis.  Otherwise, this manuscript is only a summarized work done by other researchers.

Author response: We appreciate the reviewer’s comments on strengthening the conclusion, in particular the suggestion to provide guidance for addressing disparities. Focusing on areas of healthcare provision, education, and social determinants of health, we have included examples of actionable areas and initiatives to address modifiable aspects, while at the same time acknowledging that the challenges of racial and ethnic disparities in healthcare remain complex and multifactorial.

Reviewer 2 Report

Although OA is a common condition that increases with age with a prevalence that is generally similar across racial and ethnic groups, disparities in the treatment of OA among these patients are well documented and continue to rise and persist. This is a conclusion from the study, however other similar conditions as osteoporosis has very different disease burden caused by genetic reason with Scandinavian origin most at risk. I am therefore in doubt if there can be other reasons for the difference observed, in the current manuscript.

Author Response

Reviewer 2 (Received September 21): Although OA is a common condition that increases with age with a prevalence that is generally similar across racial and ethnic groups, disparities in the treatment of OA among these patients are well documented and continue to rise and persist. This is a conclusion from the study, however other similar conditions as osteoporosis has very different disease burden caused by genetic reason with Scandinavian origin most at risk. I am therefore in doubt if there can be other reasons for the difference observed, in the current manuscript.

Author response: We thank the reviewer for their comments. We appreciate the insights regarding osteoporosis and the biological basis for differences in disease burden, and acknowledge that biological factors may play a role in disease burden disparities in osteoarthritis. However, we feel that a detailed analysis of the genetic basis of disease presentation and course is beyond the scope of the present review.